# Gut microbiome alterations among Ghanaian children with asymptomatic malaria infections

**Amma Aboagyewa Larbi**[1]*, **Moses Etsey**[1], **Obed Brew**[2‡], **Bismark Koduah**[1],
**Rosemond Enam Mawuenyega**[1], **Emmanuel Kobla Atsu Amewu**[1],
**Nehemiah Kweku Essilfie**[1], **Solomon Wireko**[3‡], **Alexander Kwarteng**[1,4],
**Ben Adu Gyan**[5‡]

1 Department of Biochemistry and Biotechnology, College of Science, Kwame Nkrumah University, of Science and Technology, Kumasi, Ghana, 2 Department of Pharmacology, College of Health Sciences, Kwame Nkrumah University of Science and Technology, Kumasi, Ghana, 3 Kumasi Technical University, Kumasi, Ghana, 4 Kumasi Centre for Collaborative Research in Tropical Medicine, Kumasi, Ghana, 5 Noguchi Memorial Institute for Medical Research, University of Ghana, Legon, Ghana

☯ These authors contributed equally to this work.
‡ These authors also contributed equally to this work.
* ammalarbi@gmail.com

## Abstract

The human gut microbiome, consisting of bacteria, archaea, fungi, and viruses, influences various physiological processes of the body. The gut microbiome composition is shaped by factors such as diet, geography, and antibiotic use. Malaria has been a global health challenge over the years, especially in low- and middle-income countries. This study investigated how asymptomatic malaria infection altered gut microbial communities in Ghanaian children, offering insights for novel malaria control strategies. Standard aseptic phlebotomy procedures were employed to collect venous blood samples for *Plasmodium* species detection. The gut microbial community was profiled by sequencing the 16S rRNA V4 region, and sequence data were processed using the DADA2 pipeline in R. Asymptomatic malaria infections were predominantly mixed with *P. falciparum* and *P. malariae*. Microbiome analysis revealed that Firmicutes and Bacteroidetes comprised nearly 70% of the total microbial population. Asymptomatic individuals showed a decrease in Firmicutes abundance from 52.5% to 44.0% and an increase in Bacteroidetes from 34.7% to 45.6%. There was also a slight increase in the abundance of Proteobacteria from 3.0% to 4.8%. At the genus level, Prevotella_9 was the most abundant and exhibited the highest variability in the infected groups. The *Alloprevotella* and *Streptococcus* genera increased in both infected groups, but *Escherichia-Shigella* was significantly elevated in only those with mixed infections. *Faecalibacterium* significantly declined in asymptomatic malaria-infected individuals compared to healthy controls, with variability further reduced in mixed infections. Beta-diversity analysis indicated a significant effect of malaria status on microbial composition (PERMANOVA, $p < 0.05$), explaining approximately 19.1% of the total variation captured by a 2D Principal Component Analysis (PCA)

**Data availability statement:** All relevant data are within the manuscript and its Supporting information files.

**Funding:** This work was supported by The World Academy of Sciences (TWAS) under Grant Number 21-111 RG/BIO/AF/AC_1-FR3240319478. It was awarded to AAL for consumable supplies. The funders had no role in study design, data collection and analysis, decision to publish, or preparation of the manuscript.

**Competing interests:** The authors have declared that no competing interests exist.

projection. These findings suggest a potential link between malaria infection and gut microbiota alterations and highlight microbial shifts associated with disease status.

## Introduction

Malaria remains a major health challenge in sub-Saharan Africa, where *Plasmodium* infections are highly endemic and disproportionately affect children. Asymptomatic malaria infections play a critical role in sustaining transmission and maintaining parasite reservoirs within endemic communities [1]. These silent carriers act as hidden reservoirs that complicate malaria control and elimination efforts [1]. Beyond its direct clinical impact, malaria infection also alters host physiology, including gut microbial dysbiosis, immune and metabolic pathways [2,3]. The interaction between malaria infection and the gut microbiome has been reported to play a role in malaria pathogenesis and influence host immunity and disease severity [4]. The human gut microbiome consists of diverse microorganisms (bacteria, archaea, fungi, and viruses) that influence various physiological processes of the body [5]. The composition of the gut microbiome is shaped by factors that impact health and disease and some of these factors include diet, geography, genetics, infection, and antibiotic use [3].

The disruption of gut microbial balance has been observed in malaria infections. Shifts in the abundance of Firmicutes, Bacteroidetes, and Proteobacteria have been reported and have potential consequences for intestinal integrity and systemic inflammation [6]. The dominant microbial phyla in the gut that contribute to the metabolism of nutrients, immune homeostasis of the body, and susceptibility to other infections are Firmicutes and Bacteroidetes [7,8]. Firmicutes is a phylum involved in gut homeostasis and short-chain fatty acid production, while Proteobacteria are linked to inflammation. As a result, a decrease in Firmicutes and an increase in Proteobacteria have been noted in malaria-infected individuals, suggesting that malaria can disrupt gut microbial stability [9,10]. This emerging evidence suggests that these alterations have broader implications beyond the infection itself and can alter metabolic pathways that will have potential consequences on recovery from the infection and response to treatment.

Despite emerging evidence linking the microbiome to malaria pathogenesis, most studies have focused on symptomatic cases, leaving limited understanding of asymptomatic infections in African children [11]. Research to understand these microbiome-related changes in asymptomatic malaria cases holds the potential to provide new insights into host-pathogen interactions and pave the way for microbiome-targeted interventions to complement the existing malaria control strategies. As a result, this study aims to characterize the gut microbiome composition in Ghanaian children with asymptomatic malaria to determine microbial shifts associated with asymptomatic *Plasmodium* infections. The study hypothesise that even when children do not show malaria symptoms, the infection still affects the gut microbiome in measurable ways.

## Materials and methods

### Ethical statement

This study was conducted in accordance with the Declaration of Helsinki and approved by the Committee on Human Research, Publication, and Ethics (CHRPE) at the Kwame Nkrumah University of Science and Technology (KNUST) and Komfo Anokye Teaching Hospital, Kumasi (Approval No. CHRPE/AP/190/22). Additional authorisation was obtained from the Ahanta West Municipal Health Directorate. Written informed consent was obtained from parents or legal guardians of all participating children prior to sample collection.

### Study design and study site

A cross-sectional design was employed to recruit 113 school-aged children (6–15 years) residing in the Ahanta West Municipality, Western Region, Ghana – an area of perennial malaria transmission. Participants were prospectively recruited between December 4, 2022 and December 30, 2022 through community health screenings. Inclusion criteria required asymptomatic status (absence of fever or malaria symptoms at the time of sampling) and residence within the study area. Children who had taken antibiotics or antimalarial medication within two weeks prior to sampling, and participants found to be infected with intestinal parasites were excluded.

### Sample collection

Venous blood samples (2–3 mL) were collected aseptically into EDTA tubes using standard phlebotomy procedures for malaria diagnosis. Stool samples were obtained using sterile containers prefilled with 2 mL RNAlater (Invitrogen, USA) to stabilise microbial RNA. Each sample was labelled with a unique identifier and stored on ice during transport to the laboratory, then frozen at –20°C until processing.

### Malaria diagnosis

Thick and thin blood smears were prepared and stained with 10% Giemsa for parasite detection and quantification under 100 × oil immersion microscopy. Rapid diagnostic tests (RDTs) were conducted using Bioline™ Malaria Ag P.f/Pan test kits according to manufacturer instructions. Molecular confirmation and *Plasmodium* species identification were performed using the Novaplex™ Malaria Assay (Seegene, Korea) on a Bio-Rad CFX Maestro real-time PCR system. Species-specific fluorescence signals enabled differentiation of *P. falciparum, P. malariae*, and *P. ovale*. Positive and negative controls were included in each batch.

### DNA extraction and 16S rRNA gene sequencing

Microbial DNA was extracted from stool samples using the QIAamp DNA Stool Mini Kit (Qiagen, Germany) following the manufacturer's instructions, with extraction blanks included as negative controls. DNA yield and purity were assessed using a NanoDrop spectrophotometer and Qubit 3.0 fluorometer. The V4 hypervariable region of the bacterial 16S rRNA gene was amplified using universal primers 515F/806R and Platinum Hot Start PCR Master Mix (Thermo Fisher Scientific). PCR products were purified, normalised, and pooled equimolarly prior to sequencing on an Illumina MiSeq platform (MiSeq Reagent Kit v3; capable of 2 × 300 bp paired-end reads), with our run generating up to 2 × 270 bp paired-end reads. Library quality was confirmed via capillary electrophoresis, and PhiX (15–20%) was spiked in to improve base diversity.

### Bioinformatics and statistical analysis

Demultiplexed paired-end reads were processed in R (v4.3.3) using the DADA2 pipeline for quality filtering, error correction, merging, and chimera removal, generating an amplicon sequence variant (ASV) table. Taxonomic classification was

performed using the SILVA v138 reference database and the naïve Bayesian classifier. The resulting ASV, taxonomy, and metadata tables were integrated in Phyloseq for diversity analyses. Low-prevalence ASVs were filtered (<0.01% of total reads). Alpha diversity (Observed Richness and Shannon index) was computed after rarefaction, and between-group differences were tested using the Wilcoxon rank-sum test. Beta diversity was assessed using Aitchison distance and visualised via Principal Component Analysis (PCA), with statistical significance evaluated using PERMANOVA (999 permutations).

## Results

### Demographic characteristics of participants

One hundred and thirteen (113) children were recruited for the study, with more than half being females (61.9%). The median age was 12 years, with a median height of 140.2 cm and a median weight of 32.1 kg. The distribution of the demographics of the study participants between asymptomatic malaria positive and negative is presented in Table 1. There was no significant difference between the two groups.

### Distribution of asymptomatic malaria infections

Out of the participants, 86 (76.11%) were confirmed to have the malaria parasite without showing any clinical symptoms. From Fig 1, mixed infections were more prevalent among the study participants, with the majority harbouring both *P. falciparum* and *P. malariae*.

### Microbiome dynamics in asymptomatic malaria patients (*P. falciparum* only)

A total of 3,497,258 reads were obtained across 37 samples, with a median of 89,732 reads per sample. After quality filtering and chimera removal, a total of 1,037 unique ASVs were retained for downstream microbiome analysis. The per-ASV read counts ranged from 4 to 446,320, with a median of 332 reads per ASV (S1 Appendix).

Table 1. Baseline characteristics of study participants.

| Variable | Negative (n=27) Mean±SD | Positive (n=86) Mean±SD | p-value |
|---|---|---|---|
| Age (years) | 12.04±2.12 | 12.12±2.37 | 0.8771 |
| Height (cm) | 141.93±13.31 | 142.23±13.35 | 0.9179 |
| Weight (kg) | 34.21±9.90 | 34.25±10.16 | 0.9856 |
| BMI-for-age Z-score (BAZ) | −0.98±0.92 | −1.03±0.87 | 0.83 |

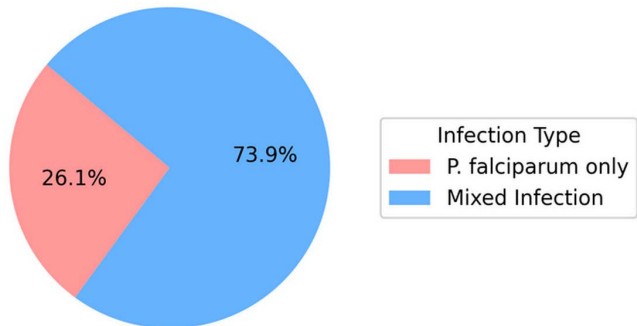

**Fig 1. Distribution of asymptomatic malaria infections.**

## Relative abundance of microbial taxa

Figs 2 and 3 illustrate the abundance of the microbial taxa and distribution at the phylum and genus level for individuals infected with *P. falciparum* only relative to healthy group. The dominant bacterial phyla identified were Firmicutes and Bacteroidetes accounting for roughly 70% of total taxa. Compared with healthy participants, the *P. falciparum*-only group exhibited a decrease in Firmicutes from 51.296% to 33.455% and an increase in Bacteroidetes from 36.646% to 58.169%. *Prevotella*_9 was the most abundant genus, followed by *Faecalibacterium* and *Blautia*. Compared with healthy controls, asymptomatic malaria cases showed a reduction in *Faecalibacterium* and increases in *Streptococcus.* All these changes were not significant but there was a notable increase in the abundance of *Alloprevotella* from 1.777% to 4.994% at a significant level of 0.036. This is an indication of microbial imbalance marked by the loss of beneficial butyrate producers and enrichment of pro-inflammatory taxa. The phylum-level and the genus-level comparison are shown in S1 Table and S2 Table respectively.

## Microbial diversity

Shannon diversity index analysis was performed to assess and compare microbial species richness and evenness between healthy individuals and those with asymptomatic *P. falciparum*-only infections, Fig 4. Although the difference

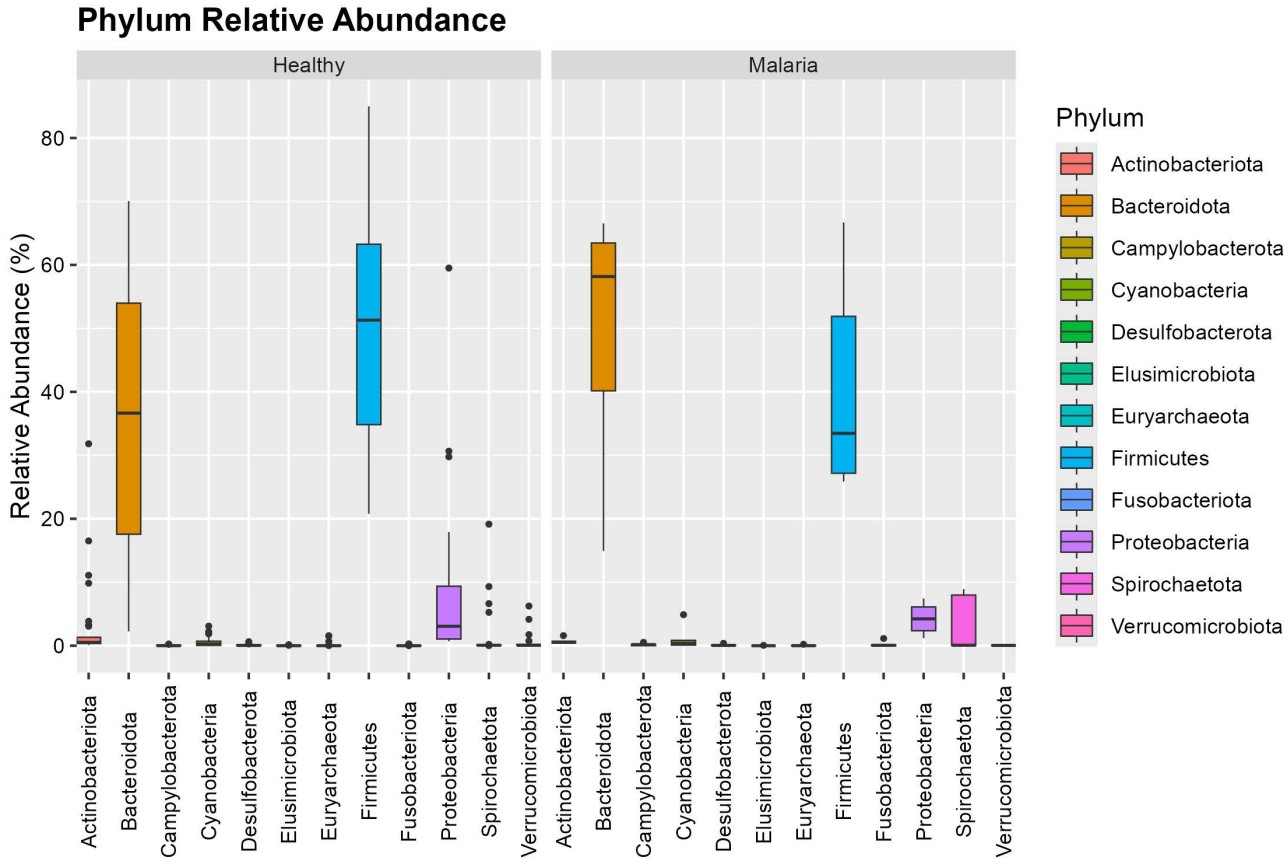

**Fig 2. Relative abundance of microbial phyla in asymptomatic *P. falciparum*-only infection.**

## Genus Relative Abundance

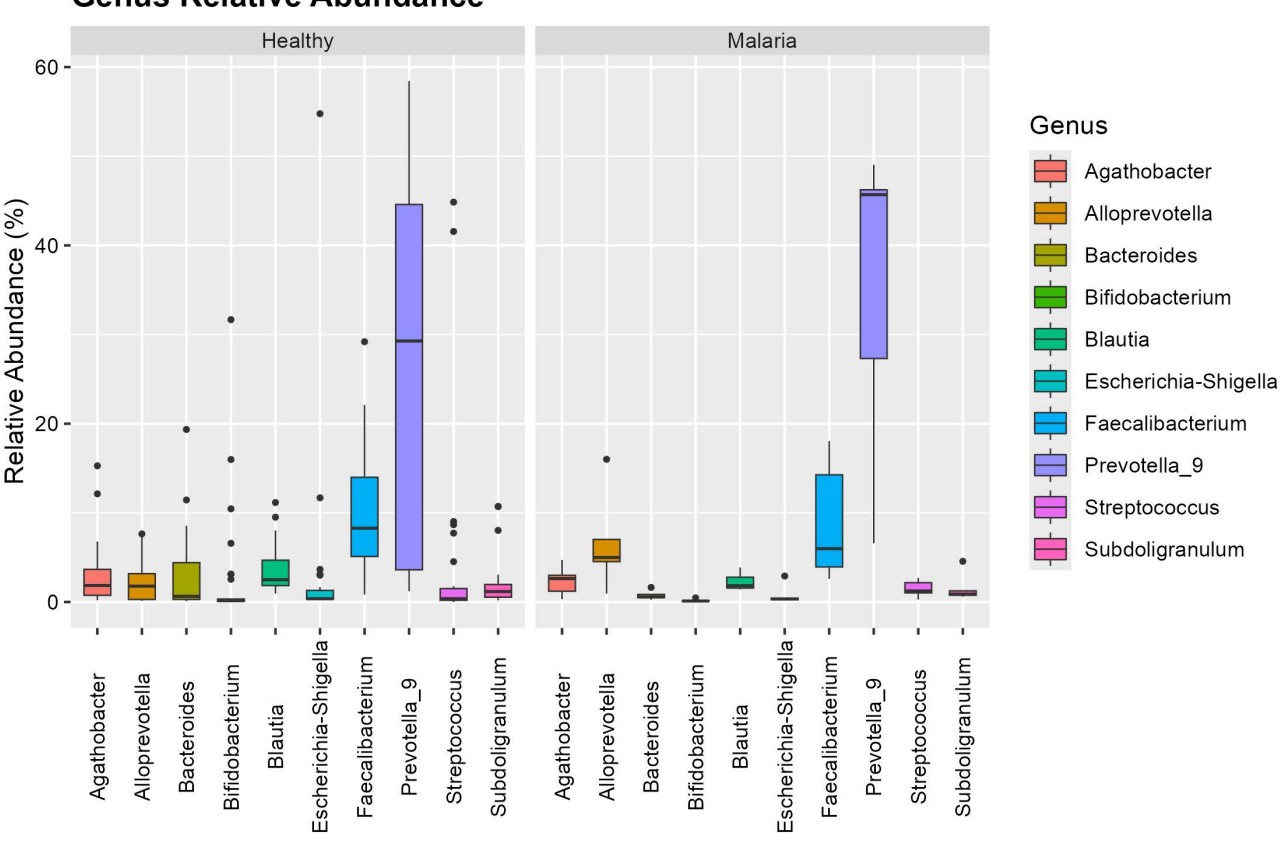

**Fig 3. Genus-level distribution in asymptomatic *P. falciparum*-only infection.**

indicates a marginal increase in microbial diversity among malaria cases, it was not statistically significant (p > 0.05). The substantial overlap in diversity ranges rather suggests that overall gut microbial richness and evenness remain largely stable despite infection. The presence of outliers in both groups further highlights pronounced inter-individual variability in gut microbial composition. The malaria group exhibited a slightly higher median Shannon index (3.8, IQR: 3.5–4.1) compared with the healthy group (3.6, IQR: 3.2–4.0), although the difference was not statistically significant (p > 0.05). The substantial overlap in interquartile ranges suggests that overall gut microbial richness and evenness remained largely stable despite infection.

### Microbiome dynamics in asymptomatic malaria patients (Mixed *plasmodium* infection)

A total of 4,348,533 high-quality reads were obtained across 48 fecal samples, with a median of 86,878 reads per sample. After quality filtering and chimera removal, 917 unique ASVs were retained for downstream analyses. Per-ASV read counts ranged from 10 to 550,286, with a median of 559 reads per ASV (S2 Appendix).

### Relative abundance of microbial taxa

The phylum-level composition in Fig 5 mirrored that of the single-infection group, Fig 2, with Firmicutes and Bacteroidetes remaining dominant. However, subtle shifts in relative abundance were observed, particularly a modest decrease in

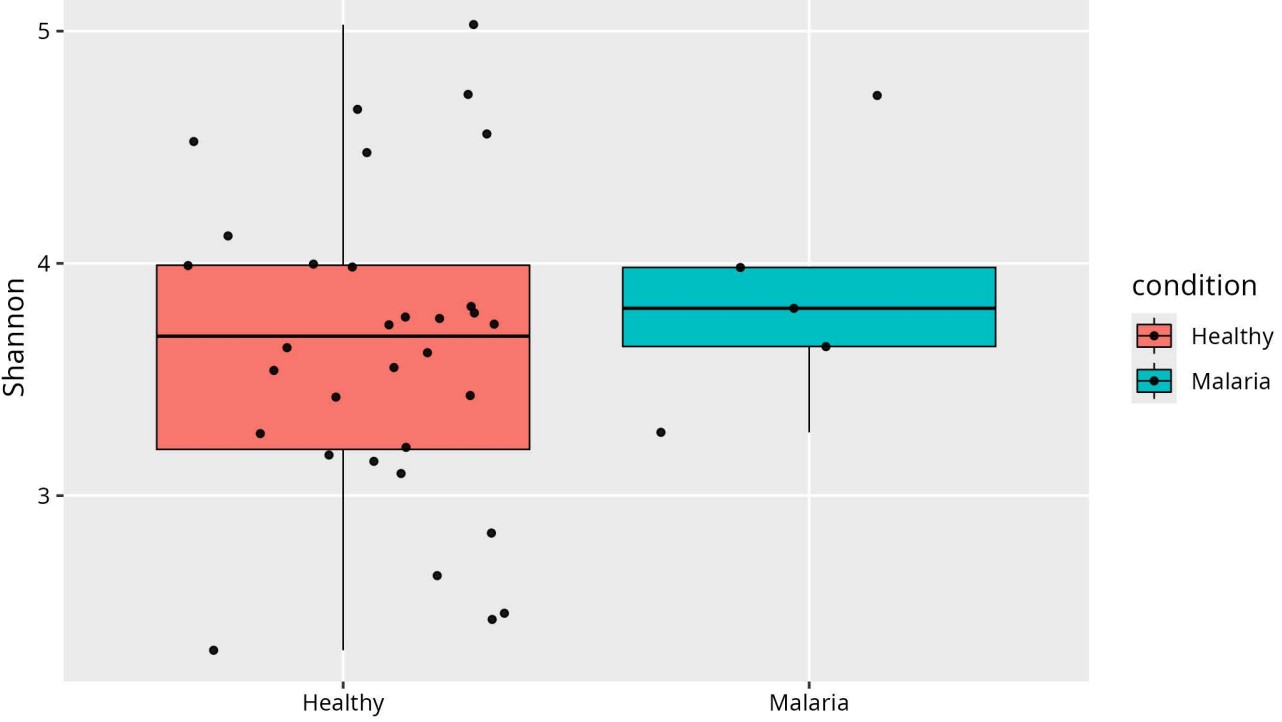

**Fig 4. Shannon diversity index comparing healthy and *P. falciparum*-only infections.**

Bacteroidetes and a compensatory increase in Firmicutes. The abundance of Spirochaetota remained relatively constant but showed reduced variability. The phylum-level comparison is shown in S3 Table and S5 Table.

Fig 6 shows that the *Prevotella_9* genus exhibited the highest variability and was the most abundant in both groups, with an increase in the infected group. The abundance of *Alloprevotella significantly* increased in asymptomatic malaria-infected individuals with mixed infection compared to healthy controls. *Faecalibacterium* was the second most abundant genus in the healthy group but showed a decline in the asymptomatic malaria-infected group, with reduced variability. Its variability was also lower compared to individuals with *P. falciparum* only. Compared to asymptomatic malaria-infected individuals with *P. falciparum* only, the mixed infection group exhibited a reduction in *Prevotella_9* and *Streptococcus*, while *Escherichia-Shigella* showed a significant increase (p = 0.0477) in abundance from 0.3475% to 1.751%. These variations suggest that mixed-species *Plasmodium* infections exert distinct and mild selective pressures on gut microbial communities. The genus-level comparison is shown in S4 Table and S6 Table

## Microbial diversity

Fig 7 presents the Shannon diversity index comparing healthy individuals with those harbouring mixed *P. falciparum* and *P. malariae* infections. The malaria group exhibited a slightly higher median Shannon index (3.7, IQR: 3.4–4.1) than the healthy group (3.6, IQR: 3.2–4.0). The small difference in Shannon index was not statistically significant (Wilcoxon rank-sum test, p > 0.05), indicating that the overall within-sample microbial diversity was broadly comparable between the two groups.

## Beta diversity

Between study group differences was determined using beta-diversity depicted with the Principal Component Analysis (PCA) plot, Fig 8. There was partial separation between malaria and healthy participants. Malaria samples clustered

## Phylum Relative Abundance

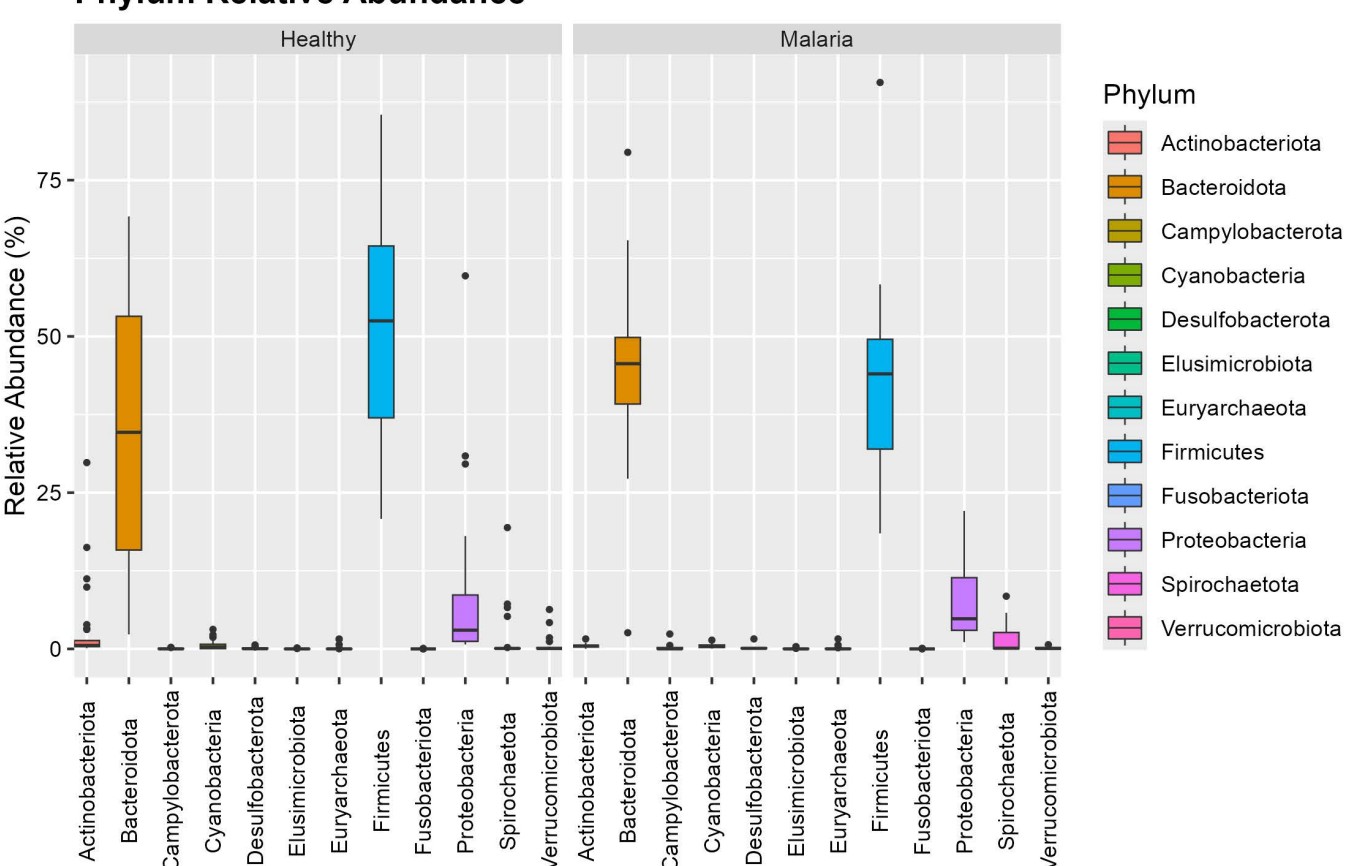

**Fig 5. Relative abundance of bacterial phyla in mixed *P. falciparum* and *P. malariae* infections.**

more closely and indicates a more uniform biological profiles associated with infection. The healthy individuals displayed greater variability. Although some overlap occurred between groups, a general trend of divergence along PC1 suggests malaria-related shifts.

Ellipses represent 95% confidence regions for each group.

## Discussion

Asymptomatic *Plasmodium* reservoirs perpetuate the transmission of malaria, making malaria control and eradication strategies virtually impossible [12]. This study focused on how this "cluster of silent parasites" affected microbiome dynamics in children in the Ahanta West, Ghana. The results provide insights into how *Plasmodium falciparum* only and mixed (*P. falciparum* and *P. malariae*) infections influence gut microbial composition and diversity. The higher prevalence of mixed infections (*P. falciparum* and *P. malariae*) compared to single *P. falciparum* infections is consistent with reports from other studies conducted in sub-Saharan Africa, where mixed-species infections are common in endemic settings [11]. According to Boundenga *et al.*, mixed infections can lead to severe malaria during the asymptomatic to symptomatic transition due to interspecies interaction, followed by increased parasitaemia. In addition, non-*falciparum* species have been found to cause severe anaemia and renal

## Genus Relative Abundance

**Fig 6. Genus-level distribution in mixed *P. falciparum* and *P. malariae* infections.**

disease in adolescents; therefore, the reservoir of mixed species in these asymptomatic carriers is a significant risk factor for complicated malaria [13].

The gut microbiome plays a central role in modulating host immunity and shaping disease outcomes through microbial–immune interactions [2,3]. Our findings thus reveal changes in microbial diversity and composition in children with asymptomatic malaria infections compared to healthy controls. Among the twelve microbial phyla identified, Firmicutes and Bacteroidetes constitute nearly 70% of the total microbial population. This distribution aligns with findings from previous microbiome studies in both healthy and malaria-infected individuals [4]. However, a shift in microbial abundance was observed such that Firmicutes, which were more abundant in healthy controls, showed a decrease in asymptomatic malaria cases, whereas Bacteroidetes increased. This shift is significant, as a reduction in Firmicutes has been associated with increased inflammation and altered gut homeostasis in animal models [4]. Thus, the observed increase in Proteobacteria suggests inflammatory stress and microbial imbalance associated with asymptomatic malaria, consistent with prior reports [9].

*Prevotella* was the most abundant taxon at the genus level, followed by *Faecalibacterium*. While *Faecalibacterium* was abundant in both groups, its reduction in asymptomatic malaria-infected individuals is of concern, as this genus is a known producer of butyrate that has anti-inflammatory properties [14]. The increase in *Alloprevotella* and *Streptococcus* in the asymptomatic malaria-infected group further highlights potential disruptions in microbial stability, as these genera have been associated with dysbiosis in other diseases [15,16].

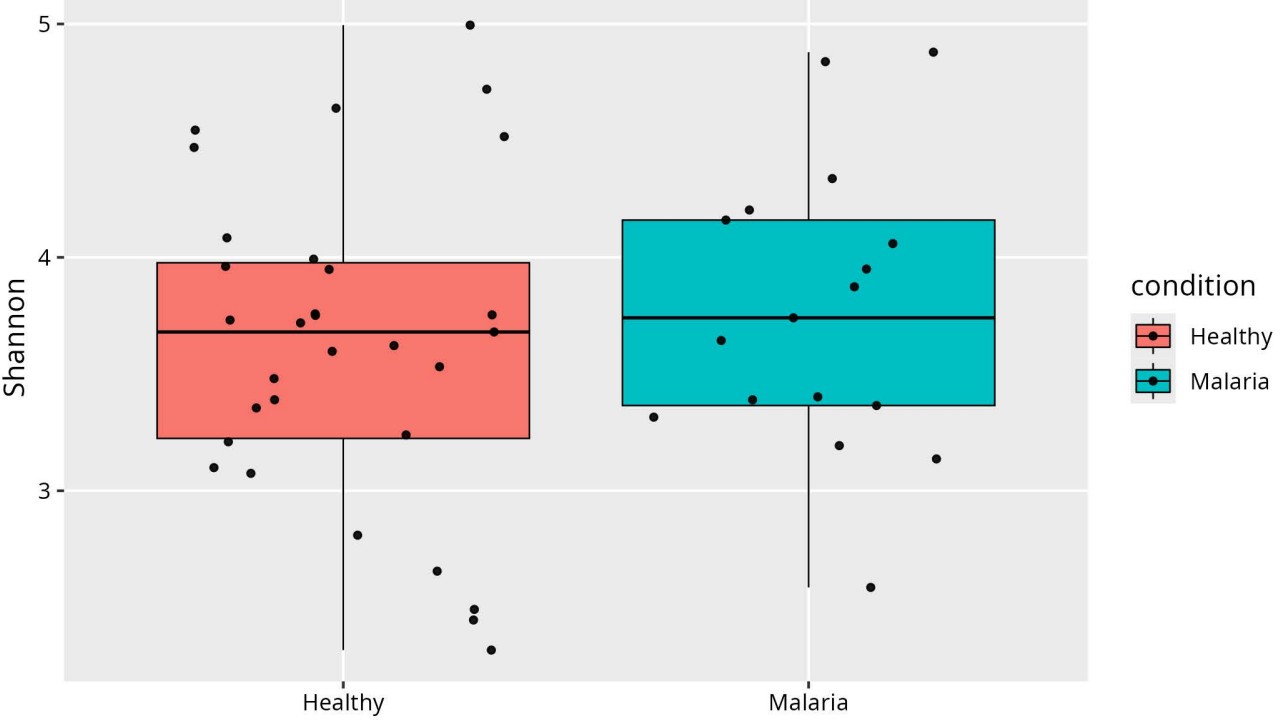

**Fig 7. Shannon diversity index comparing healthy and mixed *P. falciparum* and *P. malariae* infections.**

The children with mixed infections exhibited microbiome alterations similar to those observed in *P. falciparum*-only infections but with notable differences. The observed increase in *Escherichia-Shigella* in mixed infections is noteworthy, as this genus includes pathogenic species that contribute to gut inflammation and increased susceptibility to enteric infections [17].

Alpha-diversity analysis showed no significant differences in microbial richness and evenness between groups, suggesting that overall gut diversity remains relatively stable during asymptomatic infection. However, beta-diversity analysis revealed distinct microbial community structures associated with malaria status, indicating compositional shifts rather than broad loss of diversity. This is evident as asymptomatic malaria status accounted for more than 10% of the variation observed in the gut microbiome. These findings suggest a potential link between *Plasmodium* infection and gut microbiota alterations and highlight microbial shifts associated with disease status. While the overall diversity may not change, malaria infection significantly alters the specific composition of the gut microbiome. Future longitudinal studies that integrate metagenomics and metabolomics will be required to determine whether these compositional shifts drive susceptibility to symptomatic malaria or alter treatment response.

Diet is known to strongly influence gut microbiome composition and the absence of dietary data is a limitation for the study. Although families in the study area are generally thought to have similar dietary patterns, we cannot rule out potential dietary effects on microbial variability. Also, post-treatment microbiome data were not collected, but could provide assessment of whether the observed microbial shifts are reversible following parasite clearance. Future longitudinal studies adding dietary assessments and post-treatment sampling will be important to clarify the causal relationships between asymptomatic malaria infection and the gut microbiome alterations. This study did not include children with symptomatic malaria. Future studies comparing with symptomatic cases would provide important context and help determine whether the observed microbiome alterations are specific to asymptomatic infection.

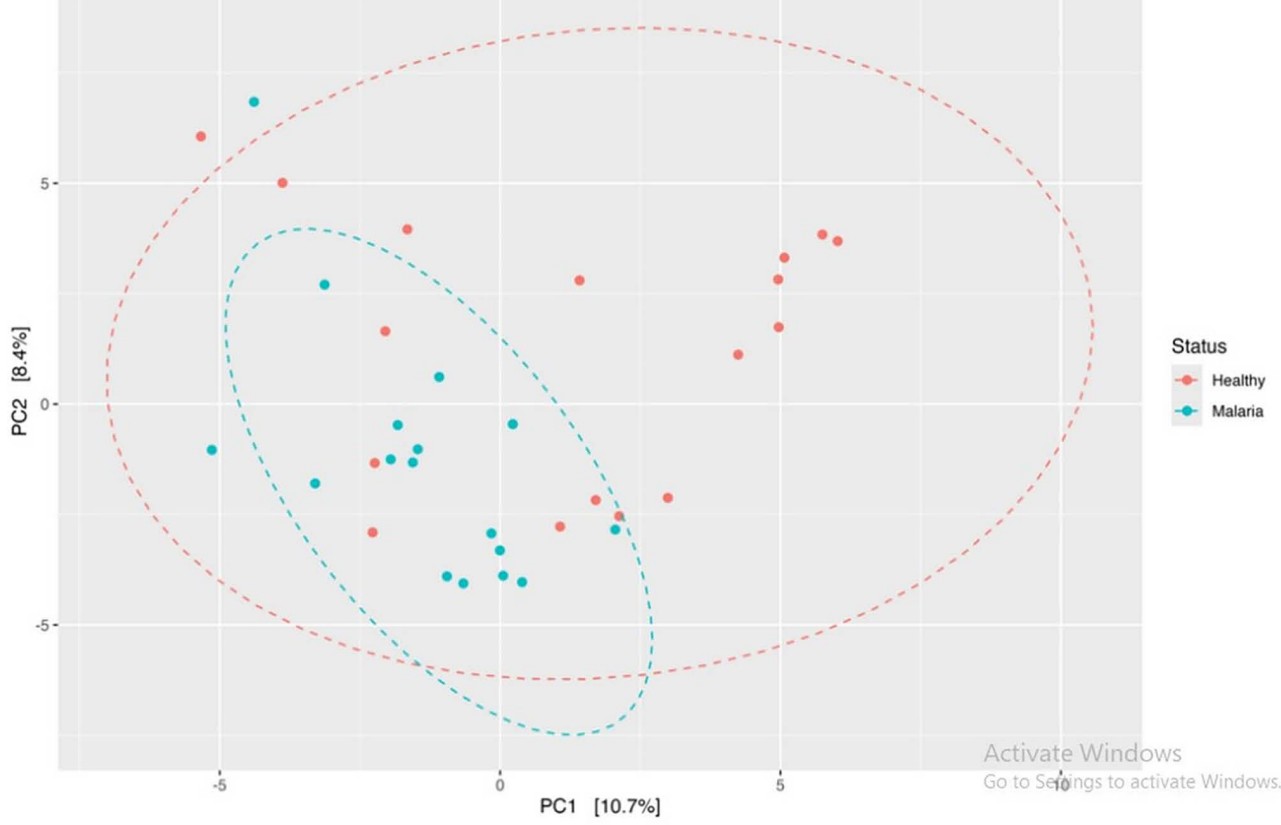

**Fig 8. PCA plot of beta-diversity.**

Another important limitation of this study is the potential inclusion of children that might have chronic conditions that were not screened for during recruitment. Although participants with known self-reported chronic conditions were excluded, screening for other conditions aside intestinal parasites infection was not performed. As a result, their potential influence on the gut microbiota composition cannot be ruled out. Future studies should consider a comprehensive screening to better control for these potential confounders.

## Conclusion

This study provides novel insights into the gut microbiota alterations in with asymptomatic malaria infections. The observed decrease in Firmicutes and *Faecalibacterium*, alongside an increase in Bacteroidetes, Proteobacteria, and *Escherichia-Shigella* is an indication of disruption in the gut microbiota. Mixed infections appear to amplify gut microbial disruption. Understanding these microbial dynamics may guide microbiome-targeted adjunct therapies to support malaria control in endemic communities.

## Supporting information

**S1 Table. Phylum-level relative abundance comparison between children with *P. falciparum*-only infection and healthy controls.**
(XLSX)

**S2 Table. Genus-level relative abundance comparison between children with *P. falciparum*-only infection and healthy controls.**
(XLSX)

**S3 Table. Phylum-level relative abundance comparison between children with mixed *Plasmodium* infection and healthy controls.**
(XLSX)

**S4 Table. Genus-level relative abundance comparison between children with mixed *Plasmodium* infection and healthy controls.**
(XLSX)

**S5 Table. Phylum-level relative abundance comparison between children with *P. falciparum*-only and mixed *Plasmodium* infection.**
(XLSX)

**S6 Table. Genus-level relative abundance comparison between children with *P. falciparum*-only and mixed *Plasmodium* infection.**
(XLSX)

**S1 Appendix. Processed phyloseq object containing ASV counts, taxonomy assignments, and sample metadata for *P. falciparum* only.**
(ZIP)

**S2 Appendix. Processed phyloseq object containing ASV counts, taxonomy assignments, and sample metadata for mixed *Plasmodium* infection.**
(ZIP)

## Acknowledgments

The authors gratefully acknowledge the microbiota vault for their services provided.

## Author contributions

**Conceptualization:** Amma Aboagyewa Larbi, Alexander Kwarteng.

**Formal analysis:** Amma Aboagyewa Larbi, Moses Etsey, Bismark Koduah, Rosemond Enam Mawuenyega.

**Funding acquisition:** Amma Aboagyewa Larbi.

**Investigation:** Amma Aboagyewa Larbi, Bismark Koduah, Rosemond Enam Mawuenyega, Emmanuel Kobla Atsu Amewu, Solomon Wireko.

**Methodology:** Amma Aboagyewa Larbi, Bismark Koduah, Rosemond Enam Mawuenyega, Emmanuel Kobla Atsu Amewu, Alexander Kwarteng.

**Project administration:** Amma Aboagyewa Larbi, Emmanuel Kobla Atsu Amewu, Alexander Kwarteng.

**Resources:** Amma Aboagyewa Larbi.

**Supervision:** Amma Aboagyewa Larbi.

**Validation:** Amma Aboagyewa Larbi, Moses Etsey, Obed Brew, Bismark Koduah, Rosemond Enam Mawuenyega, Emmanuel Kobla Atsu Amewu, Nehemiah Kweku Essilfie, Alexander Kwarteng, Ben Adu Gyan.

**Visualization:** Amma Aboagyewa Larbi, Bismark Koduah, Rosemond Enam Mawuenyega, Emmanuel Kobla Atsu Amewu.

**Writing – original draft:** Amma Aboagyewa Larbi, Moses Etsey.

**Writing – review & editing:** Amma Aboagyewa Larbi, Moses Etsey, Obed Brew, Bismark Koduah, Rosemond Enam Mawuenyega, Emmanuel Kobla Atsu Amewu, Nehemiah Kweku Essilfie, Solomon Wireko, Alexander Kwarteng, Ben Adu Gyan.

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
