## [Decision Letter · Decision Letter 0]

24 Feb 2026

Dear Dr. Larbi,

Thank you for submitting your manuscript to PLOS ONE. After careful consideration, we feel that it has merit but does not fully meet PLOS ONE’s publication criteria as it currently stands. Therefore, we invite you to submit a revised version of the manuscript that addresses the points raised during the review process.

We look forward to receiving your revised manuscript.

Kind regards,

Christopher Staley, Ph.D.

Academic Editor

PLOS One

Journal Requirements:

“AAL is supported by a grant from the World Academy of Sciences (TWAS) (Grant Number: 21-111 RG/BIO/AF/AC_1-FR3240319478). The funder had no role in study design, data collection and analysis, decision to publish, or preparation of the manuscript.”

“AAL is supported by a grant from the World Academy of Sciences (TWAS) (Grant Number: 21-111 RG/BIO/AF/AC_1-FR3240319478). The funder had no role in study design, data collection and analysis, decision to publish, or preparation of the manuscript.”

4. We note that your Data Availability Statement is currently as follows: “All relevant data are within the manuscript and its Supporting Information files.”

Additional Editor Comments:

The authors present a timely study investigating how malarial infections altered the gut microbiome. In addition to suggestions from the reviewers, the authors should include more rigorous statistical analyses of changes in taxa abundance.

Specific comments:

line 28: Details of bioinformatics processing do not need to be included in the abstract. Please remove.

line 29 and throughout: Please ensure appropriate italicization of genus and species names.

line 31 and throughout the abstract: What was the magnitude of change? Was it statistically significant?

line 39: Please define PCA at first use.

line 39: "This confirms..." this sentence is redundant with the following and the data preceding. Suggest removing.

keywords: Suggest removing and replacing words already in the title to help with indexing and searchability.

line 115: The v3 Illumina kit has a read length of 2 x 300 nt. Please clarify.

lines 146-153; 186-192: What is the utility of presenting prevalence of phyla? This doesn't seem to add value. Suggest removing. line 152 needs statistical support or a clear representation which is not currently presented.

Figure 2: Please be consistent with use of old taxonomy (e.g. Proteobacteria) vs. new taxonomy (Pseudomonadota) names. These are currently mixed.

lines 158-160: Sentence fragment.

line 160-164; 196-215: Please support with statistics.

Reviewers' comments:

Reviewer's Responses to Questions

**Comments to the Author**

1. Is the manuscript technically sound, and do the data support the conclusions?

Reviewer #1: Partly

Reviewer #2: Partly

2. Has the statistical analysis been performed appropriately and rigorously?

Reviewer #1: No

Reviewer #2: Yes

3. Have the authors made all data underlying the findings in their manuscript fully available?

Reviewer #1: Yes

Reviewer #2: No

4. Is the manuscript presented in an intelligible fashion and written in standard English?

Reviewer #1: No

Reviewer #2: Yes

Reviewer #1: The manuscript addresses a relevant topic and presents useful data. However, revision is required to improve flow and clarity. Scientific names are inconsistently italicised, the results section needs restructuring, and the discussion should better focus on interpreting the findings. Please see my comments in the manuscript

Reviewer #2: In overall, the study is well-conceived, addressing an important and timely topic. The manuscript provides valuable information on alterations to the gut microbiome in children with asymptomatic malaria, and adds to the growing body of evidence suggesting that so-called 'asymptomatic' malaria may not actually be asymptomatic. The observed microbiome changes support the idea that these infections could have biological, and potentially clinical, consequences for the children affected.

This is particularly relevant because asymptomatic malaria is often viewed primarily as a reservoir that contributes to transmission at the population level. However, if measurable physiological alterations are present, this reinforces the importance of identifying and treating these infections as a public health control strategy and an individual health priority. Therefore , the manuscript's conclusions should emphasise this concept more strongly and align better with the data presented.

Below are several clarifications and requested revisions.

Major comments

In general, it is important to note that the article does not indicate how many of the 113 recruited children have asymptomatic malaria and how many do not. This information is fundamental for evaluating the comparison. There is no table comparing the two groups and their epidemiological characteristics. Providing only the overall epidemiological variables shown in Table 1 is insufficient.

1. Distribution of other chronic infections

The manuscript does not report on the prevalence of other infections that could have a significant influence on chronic inflammatory status and the composition of the gut microbiota, such as HIV infection or hepatotropic viruses. These are important potential confounders. The distribution of such infections across groups should be reported and controlled for in the analysis.

2. Nutritional status of participants

Nutritional status is a major determinant of gut microbiome composition and immune function. The authors should provide anthropometric data, at least BMI. It should also be clarified whether the distribution of BMI differed between groups. The reported mean BMI of 16.3 suggests that many children may be underweight. A more detailed characterisation of nutritional status is necessary to determine whether the groups are truly comparable.

3. Dietary assessment

Dietary intake strongly influences microbiome composition. The absence of dietary data is a significant limitation. If such data were collected, they should be included. Otherwise, this limitation should be clearly acknowledged and discussed.

4.Intestinal parasites

The authors should explain why children infected with intestinal parasites were not excluded from the study. Such infections can directly affect the composition of the gut microbiota and may distort the results.

5.Post-treatment microbiome assessment

A post-treatment microbiome analysis would strengthen causal inference by demonstrating whether microbiome alterations are reversed after parasite clearance. While this may not be feasible retrospectively, this should be discussed as a limitation.

6. Comparisons with symptomatic malaria

Including or discussing comparisons with children with symptomatic malaria would have provided important context and helped determine whether the observed alterations are specific to asymptomatic infection.

Conclusion

In summary, this manuscript addresses an important question and provides data that potentially support the concept that asymptomatic malaria infections have biological consequences. However, several major methodological and interpretative issues must be addressed before the manuscript can be considered for publication. Strengthening the characterisation of the study population, clarifying potential confounders, moderating speculative claims and improving figure quality would substantially enhance the rigour and impact of the work.

.

Reviewer #1: No

Reviewer #2: No

---

## [Author Response · Author response to Decision Letter 1]

6 Mar 2026

Editor Comments

Response:

Thank you for the opportunity to revise our manuscript. We appreciate the constructive comments provided.

We have revised the manuscript and addressed all comments raised. A point-by-point response to the reviewers has been provided. In addition, we have updated the Funding Statement, and all raw data required to replicate the results of our study have been added as Supporting Information.

Reviewer #1

Comment:

The manuscript addresses a relevant topic and presents useful data. However, revision is required to improve flow and clarity. Scientific names are inconsistently italicised, the results section needs restructuring, and the discussion should better focus on interpreting the findings. Please see my comments in the manuscript.

Response:

We have revised the manuscript with scientific names now consistently italicised throughout the text. The Results section has been reorganised for improved logical progression. All specific comments in the manuscript have been addressed accordingly.

Reviewer #2

In general, it is important to note that the article does not indicate how many of the 113 recruited children have asymptomatic malaria and how many do not. This information is fundamental for evaluating the comparison. There is no table comparing the two groups and their epidemiological characteristics. Providing only the overall epidemiological variables shown in Table 1 is insufficient.

Response:

We have now clearly stated the proportion of children with asymptomatic malaria (PCR-positive, n = 86) and those without infection (PCR-negative, n = 27) in the Results section. In addition, we have included a new table comparing key demographic characteristics (age, height, weight, and BMI-for-age Z-scores) between the two groups, along with corresponding statistical significant levels.

Comment 1:

Distribution of other chronic infections

The manuscript does not report on the prevalence of other infections that could have a significant influence on chronic inflammatory status and the composition of the gut microbiota, such as HIV infection or hepatotropic viruses. These are important potential confounders. The distribution of such infections across groups should be reported and controlled for in the analysis.

Response:

Participants with known self-reported chronic infections were excluded from the microbiome analysis to minimize potential confounding effects. This has now been added to the Methods section.

Comment 2:

Nutritional status of participants

Nutritional status is a major determinant of gut microbiome composition and immune function. The authors should provide anthropometric data, at least BMI. It should also be clarified whether the distribution of BMI differed between groups. The reported mean BMI of 16.3 suggests that many children may be underweight. A more detailed characterisation of nutritional status is necessary to determine whether the groups are truly comparable.

Response:

BMI-for-age Z-scores (BMIA-Z) has been calculated for all participants. The mean BMIA-Z was −0.98 ± 0.92 in PCR-negative children and −1.03 ± 0.87 in PCR-positive children, with no significant difference between groups. These are now in the Results section in characteristics table.

Comment 3:

Dietary assessment

Dietary intake strongly influences microbiome composition. The absence of dietary data is a significant limitation. If such data were collected, they should be included. Otherwise, this limitation should be clearly acknowledged and discussed.

Response:

We agree that dietary intake strongly influences gut microbiome composition. Dietary data were not collected in this study, which limits our ability to assess its contribution. We have now explicitly acknowledged this limitation in the Discussion section.

Comment 4:

Intestinal parasites

The authors should explain why children infected with intestinal parasites were not excluded from the study. Such infections can directly affect the composition of the gut microbiota and may distort the results.

Response:

All children were screened for intestinal parasites. Participants found to be infected were excluded from the microbiome analysis. It has been added as exclusion criterion in the Methods section.

Comment 5:

Post-treatment microbiome assessment

A post-treatment microbiome analysis would strengthen causal inference by demonstrating whether microbiome alterations are reversed after parasite clearance. While this may not be feasible retrospectively, this should be discussed as a limitation

Response:

We agree that longitudinal follow-up after parasite clearance would strengthen causal inference. However, such sampling was not collected for this study. We have acknowledged this as a limitation in the Discussion and suggested it as an important direction for future research.

Comment 6:

Comparisons with symptomatic malaria

Including or discussing comparisons with children with symptomatic malaria would have provided important context and helped determine whether the observed alterations are specific to asymptomatic infection.

Response:

Our study recruited children with asymptomatic malaria, and symptomatic cases were not included in this cohort. We have now acknowledged this as a limitation in the Discussion section and noted that inclusion of symptomatic cases in future studies would help clarify whether observed microbiome alterations are specific to asymptomatic infection.

---

## [Decision Letter · Decision Letter 1]

29 Mar 2026

Dear Dr. Larbi,

Thank you for submitting your manuscript to PLOS ONE. After careful consideration, we feel that it has merit but does not fully meet PLOS ONE’s publication criteria as it currently stands. Therefore, we invite you to submit a revised version of the manuscript that addresses the points raised during the review process.

As the corresponding author, your ORCID iD is verified in the submission system and will appear in the published article. PLOS supports the use of ORCID, and we encourage all coauthors to register for an ORCID iD and use it as well. Please encourage your coauthors to verify their ORCID iD within the submission system before final acceptance, as unverified ORCID iDs will not appear in the published article. *Only* the individual author can complete the verification step; PLOS staff the individual author can complete the verification step; PLOS staff the individual author can complete the verification step; PLOS staff the individual author can complete the verification step; PLOS staff *cannot* verify ORCID iDs on behalf of authors.verify ORCID iDs on behalf of authors.verify ORCID iDs on behalf of authors.verify ORCID iDs on behalf of authors.

We look forward to receiving your revised manuscript.

Kind regards,

Christopher Staley, Ph.D.

Academic Editor

PLOS One

Journal Requirements:

Reviewers' comments:

Reviewer's Responses to Questions

**Comments to the Author**

Reviewer #1: All comments have been addressed

Reviewer #2: (No Response)

2. Is the manuscript technically sound, and do the data support the conclusions?

Reviewer #1: Yes

Reviewer #2: Yes

3. Has the statistical analysis been performed appropriately and rigorously?

Reviewer #1: Yes

Reviewer #2: Yes

4. Have the authors made all data underlying the findings in their manuscript fully available?

Reviewer #1: Yes

Reviewer #2: (No Response)

5. Is the manuscript presented in an intelligible fashion and written in standard English?

Reviewer #1: Yes

Reviewer #2: Yes

Reviewer #1: (No Response)

Reviewer #2: The authors have adequately addressed the points raised previously. Any issues related to the study design that cannot be modified have now been appropriately acknowledged in the 'Methods' and 'Discussion' sections.

However, one important aspect remains insufficiently addressed. Specifically, the potential inclusion of children with chronic diseases that may influence systemic inflammation and consequently the microbiota has not been clearly specified. Furthermore, the absence of a screening process for such conditions, particularly those that may be asymptomatic or undiagnosed, is a limitation as their potential impact cannot be ruled out. This should be explicitly acknowledged in the discussion as it is relevant to the interpretation of the results and constitutes an important limitation of the study.

.

Reviewer #1: No

Reviewer #2: No

---

## [Author Response · Author response to Decision Letter 2]

1 Apr 2026

We appreciate this important comment to address the potential inclusion of children with chronic diseases that may influence systemic inflammation and consequently the microbiota. Participants with known self-reported chronic conditions were excluded. However, we acknowledge that this approach may have not fully accounted for undiagnosed or asymptomatic chronic conditions. We have now included this as a limitation in the Discussion section as recommended.

---

## [Editor Report · Decision Letter 2]

12 Apr 2026

Gut microbiome alterations among Ghanaian children with asymptomatic malaria infections

PONE-D-26-02351R2

Dear Dr. Larbi,

We’re pleased to inform you that your manuscript has been judged scientifically suitable for publication and will be formally accepted for publication once it meets all outstanding technical requirements.

Kind regards,

Christopher Staley, Ph.D.

Academic Editor

PLOS One
---

## [Editor Report · Acceptance letter]

PONE-D-26-02351R2

PLOS One

Dear Dr. Larbi,

I'm pleased to inform you that your manuscript has been deemed suitable for publication in PLOS One. Congratulations! Your manuscript is now being handed over to our production team.

Kind regards,

on behalf of

Dr. Christopher Staley

Academic Editor

PLOS One